# A Large Batch Optimizer Reality Check: Traditional, Generic Optimizers Suffice Across Batch Sizes

## Abstract

Recently the LARS and LAMB optimizers have been proposed for training neural networks faster using large batch sizes. LARS and LAMB add layer-wise normalization to the update rules of Heavy-ball momentum and Adam, respectively, and have become popular in prominent benchmarks and deep learning libraries. However, without fair comparisons to standard optimizers, it remains an open question whether LARS and LAMB have any benefit over traditional, generic algorithms. In this work we demonstrate that standard optimization algorithms such as Nesterov momentum and Adam can match or exceed the results of LARS and LAMB at large batch sizes. Our results establish new, stronger baselines for future comparisons at these batch sizes and shed light on the difficulties of comparing optimizers for neural network training more generally.

## 1 Introduction

In recent years, hardware systems employing GPUs and TPUs have enabled neural network training programs to process dramatically more data in parallel than ever before. The most popular way to exploit these systems is to increase the batch size in the optimization algorithm (i.e. the number of training examples processed per training step). On many workloads, modern systems can scale to larger batch sizes without significantly increasing the time per step (Jouppi et al., 2017; Wang et al., 2019), thus proportionally increasing the number of training examples processed per second. If researchers can use this increased throughput to reduce the time required to train each neural network, then they should achieve better results by training larger models, using larger datasets, and by exploring new ideas more rapidly.

As the capacity for data parallelism continues to increase, practitioners can take their existing, well-tuned training configurations and re-train with larger batch sizes, hoping to achieve the same performance in less training time (e.g. Ying et al., 2018). On an idealized data-parallel system with negligible overhead from increasing the batch size, they might hope to achieve *perfect scaling*, a proportional reduction in training time as the batch size increases.

However, achieving perfect scaling is not always straightforward. Changing the batch size changes the training dynamics, requiring the training hyperparameters (e.g. learning rate) to be carefully re-tuned in order to maintain the same level of validation performance.[1] In addition, smaller batch sizes provide implicit regularization from gradient noise (Smith et al., 2021) that may need to be replaced by other forms of regularization when the batch size is increased. Finally, even with perfect tuning, increasing the batch size eventually produces diminishing returns. After a critical batch size, the number of training steps cannot be decreased in proportion to the batch size – the number of epochs must increase to match the validation performance of the smaller batch size. See Shallue et al. 2019 for a survey of the effects of data parallelism on neural network training. Once these effects are taken into account, there is no strong evidence that increasing the batch size degrades the maximum achievable performance on any workload. At the same time, the ever-increasing capacity for data parallelism presents opportunities for new regularization techniques that can replace the gradient

---

[1] Although there are heuristics for adjusting the learning rate as the batch size changes, these heuristics inevitably break down sufficiently far from the initial batch size and it is also not clear how to apply them to other training hyperparameters (e.g. momentum), as detailed in Shallue et al. (2019).

noise of smaller batch sizes and new optimization algorithms that can extend perfect scaling to larger batch sizes by using more sophisticated gradient information (Zhang et al., 2019).

You et al. (2017) proposed the LARS optimization algorithm in the hope of speeding up neural network training by exploiting larger batch sizes. LARS is a variant of stochastic gradient descent (SGD) with momentum (Polyak, 1964) that applies layer-wise normalization before applying each gradient update. Although it is difficult to draw strong conclusions from the results presented in the LARS paper, [2] the MLPerf[3] Training benchmark[4] adopted LARS as one of two allowed algorithms in the closed division for ResNet-50 on ImageNet and it became the *de facto* standard algorithm for that benchmark task. With MLPerf entrants competing to find the fastest-training hyperparameters for LARS, the first place submissions in the two most recent MLPerf Training competitions used LARS to achieve record training speeds with batch sizes of 32,678 and 65,536, respectively. No publications or competitive submissions to MLPerf have attempted to match these results with a standard optimizer (e.g. Momentum or Adam). However, MLPerf entrants do not have a strong incentive (nor are necessarily permitted by the rules) to explore other algorithms because MLPerf Training is a systems benchmark that requires algorithmic equivalence between submissions to make fair comparisons. Moreover, since the main justification for LARS is its excellent performance on ResNet-50 at large batch sizes, more work is needed to quantify any benefit of LARS over standard algorithms at any batch size.

You et al. (2019) later proposed the LAMB optimizer to speed up pre-training for BERT (Devlin et al., 2018) using larger batch sizes after concluding that LARS was not effective across workloads. LAMB is a variant of Adam (Kingma & Ba, 2014) that adds a similar layer-wise normalization step to LARS. You et al. (2019) used LAMB for BERT pre-training with batch sizes up to 65,536 and claimed that Adam cannot match the performance of LAMB beyond batch size 16,384.

In this paper, we demonstrate that standard optimizers, without any layer-wise normalization techniques, can match or improve upon the large batch size results used to justify LARS and LAMB. In Section 2, we show that Nesterov momentum (Nesterov, 1983) matches the performance of LARS on the ResNet-50 benchmark with batch size 32,768. We are the first to match this result with a standard optimizer. In Section 3, contradicting the claims in You et al. (2019), we show that Adam obtains better BERT pre-training results than LAMB at the largest batch sizes, resulting in better downstream performance metrics after fine-tuning.

In addition, we establish a new state-of-the-art for BERT pretraining speed, reaching an F1 score of 90.46 in 7,818 steps using Adam at batch size 65,536 (we report training speed in steps because our focus is algorithmic efficiency, but since we compare LARS and LAMB to simpler optimizers, fewer training steps corresponds to faster wall-time in an optimized implementation – our BERT result with Adam also improves upon the wall-time record of LAMB reported in You et al. 2019). Taken together, our results establish stronger training speed baselines for these tasks and batch sizes, which we hope will assist future work aiming to accelerate training using larger batch sizes.

In addition to the contributions mentioned above, we demonstrate several key effects that are often overlooked by studies aiming to establish the superiority of new optimization algorithms. We show that future work must carefully disentangle regularization and optimization effects when comparing a new optimizer to baselines. We also report several under-documented details used to generate the best LARS and LAMB results, a reminder that future comparisons should document any novel tricks and include them in baselines. Finally, our results add to existing evidence in the literature on the difficulty of performing independently rigorous hyperparameter tuning for optimizers and baselines. In particular, we show that the optimal shape of the learning rate schedule is optimizer-dependent (in addition to the scale), and that differences in the schedule can dominate optimizer comparisons at smaller step budgets and become less important at larger step budgets.

## 1.1 RELATED WORK

Shallue et al. (2019) and Zhang et al. (2019) explored the effects of data parallelism on neural network training for different optimizers, finding no evidence that larger batch sizes degrade performance and

---

[2] The modified AlexNet on ImageNet benchmark did not have well-established accuracy targets from prior work and LARS used a more general learning rate schedule than the momentum baseline. For ResNet-50 on ImageNet, LARS achieved sub-par accuracy numbers and was not compared to any other optimizer at the same batch size, leaving open the possibility that a generic optimizer would scale just as well as LARS. [3] MLPerf is a trademark of MLCommons.org. [4] https://mlperf.org/training-overview

demonstrating that different optimizers can achieve perfect scaling up to different critical batch sizes. Geiping et al. (2021) found that full batch gradient descent can achieve competitive performance on CIFAR-10. You et al. (2017; 2019) developed the LARS and LAMB optimizers in the hope of speeding up training by achieving perfect scaling beyond standard optimizers. Many other recent papers have proposed new optimization algorithms for generic batch sizes or larger batch sizes (see Schmidt et al., 2020). Choi et al. (2019), Sivaprasad et al. (2020), and Schmidt et al. (2020) demonstrated the difficulties with fairly comparing optimizers, showing that the hyperparameter tuning protocol is a key determinant of optimizer rankings. The MLPerf Training benchmark (Mattson et al., 2019) provides a competitive ranking of neural network training systems, but does not shed much light on the relative performance of optimizers because entrants are limited in the algorithms they can use and the hyperparameters they can tune.

## 2 MATCHING LARS ON IMAGENET

The MLPerf training benchmark for ResNet-50 v1.5 on ImageNet (Mattson et al., 2019) aims to reach 75.9% validation accuracy in the shortest possible wall-clock time. In the closed division of the competition, entrants must choose between two optimizers, SGD with momentum or LARS, and are only allowed to tune a specified subset of the optimization hyperparameters, with the remaining hyperparameter values set by the competition rules.[5] The winning entries in the two most recent competitions used LARS with batch size 32,768 for 72 training epochs[6] and LARS with batch size 65,536 for 88 training epochs,[7] respectively. Kumar et al. (2019) later improved the training time for batch size 32,768 by reaching the target accuracy in 64 epochs. These are currently the fastest published results on the ResNet-50 benchmark. However, it has been unclear whether LARS was necessary to achieve these training speeds since no recent published results or competitive MLPerf submissions have used another optimizer. In this section, we describe how we matched the 64 epoch, 32,768 batch size result of LARS using standard Nesterov momentum.[8]

A fair benchmark of training algorithms or hardware systems must account for stochasticity in individual training runs. In the MLPerf competition, the benchmark metric is the mean wall-clock time of 5 trials after the fastest and slowest trials are excluded. Only 4 out of the 5 trials need to reach the target accuracy and there is no explicit limit on the number of times an entrant can try a different set of 5 trials. Since our goal is to compare algorithms, rather than systems, we aim to match the LARS result in terms of training steps instead (but since Nesterov momentum is computationally simpler than LARS, this would also correspond to faster wall-clock time on an optimized system). Specifically, we measure the median validation accuracy over 50 training runs with a fixed budget of 2,512 training steps[9] at a batch size of 32,768. When we ran the published LARS training pipeline,[10] LARS achieved a median accuracy of 75.97% and reached the target in 35 out of 50 trials. We consider the LARS result to be matched by another optimizer if the median over 50 trials exceeds the target of 75.9%.

### 2.1 NESTEROV MOMENTUM AT BATCH SIZE 32K

This section describes how we used the standard Nesterov momentum optimizer to train the ResNet-50 v1.5 on ImageNet to 75.9% validation accuracy in 2,512 update steps at a batch size of 32,768, matching the best published LARS result at this batch size. Although we implemented our own training program, the only logical changes we made to the published LARS pipeline were to the optimizer and the optimization hyperparameters. Our model implementation and data pre-processing pipeline were identical to those required under the MLPerf closed division rules (see Appendix B).

We present two Nesterov momentum hyperparameter configurations that achieve comparable performance to LARS. Configuration A achieved a median accuracy of 75.97% (the same as LARS) and reached the target accuracy in 34 out of 50 trials. Configuration B is a modified version of Configuration A designed to make as few changes as possible to the LARS hyperparameters; it

---

[5] https://git.io/JtknD  [6] https://mlperf.org/training-results-0-6

[7] https://mlperf.org/training-results-0-7  [8] The 88 epoch, 65,536 batch size result is faster in terms of wall-clock time but requires more training epochs, indicating that it is beyond LARS's perfect scaling regime. Although LARS obtains diminishing returns when increasing the batch size from 32,768 to 65,536, future work could investigate whether Nesterov momentum drops off more or less rapidly than LARS.

[9] Corresponding to 64 training epochs in Kumar et al. (2019).  [10] https://git.io/JtsLQ

achieved a median accuracy of 75.92% and reached the target in 29 out of 50 trials. See Appendix E.1 for the complete hyperparameter configurations.

To achieve these results, we tuned the hyperparameters of the training pipeline from scratch using Nesterov momentum. We ran a series of experiments, each of which searched over a hand-designed hyperparameter search space using quasi-random search (Bousquet et al., 2017). Between each experiment, we modified the previous search space and/or tweaked the training program to include optimization tricks and non-default hyperparameter values we discovered in the state-of-the-art LARS pipeline. The full sequence of experiments we ran, including the number of trials, hyperparameters tuned, and search space ranges, are provided in Appendix E.4. Once we had matched the LARS result with Configuration A, we tried setting each hyperparameter to its value in the LARS pipeline in order to find the minimal set of changes that still achieved the target result, producing Configuration B. The remainder of this section describes the hyperparameters we tuned and the techniques we applied on the journey to these results.

### 2.1.1 NESTEROV MOMENTUM OPTIMIZER

Nesterov momentum is a variant of classical or "heavy-ball" momentum defined by the update rule

$$v_{t+1} = \mu v_t + \nabla\ell(\theta_t),$$
$$\theta_{t+1} = \theta_t - \eta_t \left(\mu v_{t+1} + \nabla\ell(\theta_t)\right),$$

where $v_0 = 0$, $\theta_t$ is the vector of model parameters after $t$ steps, $\nabla\ell(\theta_t)$ is the gradient of the loss function $\ell(\theta)$ averaged over a batch of training examples, $\mu$ is the momentum, and $\eta_t$ is the learning rate for step $t$. We prefer Nesterov momentum over classical momentum because it tolerates larger values of its momentum parameter (Sutskever et al., 2013) and sometimes outperforms classical momentum, although the two algorithms perform similarly on many tasks (Shallue et al., 2019; Choi et al., 2019). We tuned the Nesterov momentum $\mu$ in Configurations A and B. We discuss the learning rate schedule $\{\eta_t\}$ separately in Section 2.1.4.

### 2.1.2 BATCH NORMALIZATION

The ResNet-50 v1.5 model uses batch normalization (Ioffe & Szegedy, 2015), defined as

$$\texttt{BN}(x^{(l)}) = \left(\frac{x^{(l)} - \texttt{mean}(x^{(l)})}{\sqrt{\texttt{var}(x^{(l)}) + \epsilon}}\right) \times \gamma^{(l)} + \beta^{(l)},$$

where $x^{(l)}$ is a vector of pre-normalization outputs from layer $l$, $\texttt{mean}(\cdot)$ and $\texttt{var}(\cdot)$ denote the element-wise sample mean and variance across the batch of training examples,[11] and $\gamma^{(l)}$ and $\beta^{(l)}$ are trainable model parameters.

Batch normalization introduces the following tuneable hyperparameters: $\epsilon$, the small constant added to the sample variance; the initial values of $\gamma^{(l)}$ and $\beta^{(l)}$; and $\rho$, which governs the exponential moving averages of the scaling factors used in evaluation. The LARS pipeline uses $\epsilon = 10^{-5}$ and $\rho = 0.9$. It sets the initial value of $\beta^{(l)}$ to 0.0 everywhere, but the initial value of $\gamma^{(l)}$ depends on the layer: it sets $\gamma^{(l)}$ to 0.0 in the final batch normalization layer of each residual block, and to 1.0 everywhere else. In Configuration A, we tuned $\epsilon$, $\rho$, and $\gamma_0$, the initial value of $\gamma^{(l)}$ in the final batch normalization layer of each residual block. In Configuration B, we used the same values as LARS for $\epsilon$ and $\rho$, but we found that choosing $\gamma_0$ between 0.0 and 1.0 was important for matching the LARS result with Nesterov momentum.

### 2.1.3 REGULARIZATION

In Configuration A, we tuned both the L2 regularization coefficient $\lambda$ and label smoothing coefficient $\tau$ (Szegedy et al., 2016). The LARS pipeline uses $\lambda = 10^{-4}$ and $\tau = 0.1$. Crucially, the LARS pipeline does not apply L2 regularization to the bias variables of the ResNet model nor the batch normalization parameters $\gamma^{(l)}$ and $\beta^{(l)}$ (indeed, the published LARS pipeline does not even apply LARS to these parameters – it uses Heavy-ball momentum). This detail is extremely important for both LARS and Nesterov momentum to achieve the fastest training speed. Configuration B used the same $\lambda$ and $\tau$ as Configuration A.

---

[11] In a distributed training environment the mean and variance are commonly computed over a subset of the full batch. The LARS pipeline uses a "virtual batch size" of 64, which we also use to avoid changing the training objective (Hoffer et al., 2017).

### 2.1.4 Learning rate schedule

The LARS pipeline uses a piecewise polynomial schedule

$$
\eta_t = \begin{cases} \eta_{\text{init}} + (\eta_{\text{peak}} - \eta_{\text{init}}) \left( \frac{t}{t_{\text{warmup}}} \right)^{p_{\text{warmup}}}, & t \leq t_{\text{warmup}} \\ \eta_{\text{final}} + (\eta_{\text{peak}} - \eta_{\text{final}}) \left( \frac{T-t}{T-t_{\text{warmup}}} \right)^{p_{\text{decay}}} & t > t_{\text{warmup}}, \end{cases}
$$

with $\eta_{\text{init}} = 0.0$, $\eta_{\text{peak}} = 29.0$, $\eta_{\text{final}} = 10^{-4}$, $p_{\text{warmup}} = 1$, $p_{\text{decay}} = 2$, and $t_{\text{warmup}} = 706$ steps. In Configuration A, we re-tuned all of these hyperparameters with Nesterov momentum. In Configuration B, we set $\eta_{\text{init}}$, $p_{\text{decay}}$, and $t_{\text{warmup}}$ to the same values as LARS, changing only $p_{\text{warmup}}$ from 1 to 2 and re-scaling $\eta_{\text{peak}}$ and $\eta_{\text{final}}$.

|  | Nesterov | LARS |
|---|---|---|
| $p_{\text{warmup}}$ | 2 | 1 |
| $\eta_{\text{peak}}$ | 7.05 | 29.0 |
| $\eta_{\text{final}}$ | $6 \times 10^{-6}$ | $10^{-4}$ |
| $1 - \mu$ | 0.02397 | 0.071 |
| $\lambda$ | $5.8 \times 10^{-5}$ | $10^{-4}$ |
| $\tau$ | 0.15 | 0.10 |
| $\gamma_0$ | 0.4138 | 0.0 |

Table 1: The hyperparameters of Configuration B that differ from state-of-the-art LARS at batch size 32,768 (Kumar et al., 2019).

### 2.1.5 Comparing Nesterov momentum and LARS

Table 1 shows the hyperparameter values for Configuration B that differ from the state-of-the-art LARS pipeline. Aside from re-tuning the momentum, learning rate scale, and regularization hyperparameters (whose optimal values are all expected to change with the optimizer), the only changes are setting $p_{\text{warmup}}$ to 2 instead of 1 and re-tuning $\gamma_0$.

Figure 1 shows the LARS learning rate schedule compared to the Nesterov momentum schedule. Even though these schedules are similar, we found that each optimizer had a different optimal value of the warmup polynomial power. As Table 2 shows, Nesterov momentum performs better with $p_{\text{warmup}} = 2$ instead of 1, while the opposite is true with LARS. As discussed in Agarwal et al. (2020), optimizers can induce implicit step size schedules that strongly influence their training dynamics and solution quality (Li et al., 2019), and it appears from Table 2 that the implicit step sizes of Nesterov momentum and LARS may evolve differently, causing the shapes of their optimal learning rate schedules to differ.

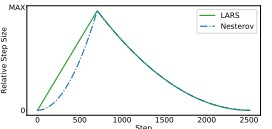

Figure 1: The learning rate schedules of LARS and Nesterov momentum Configuration B. Aside from re-scaling, the only difference is setting the warmup polynomial power to 2 instead of 1.

Although the main concern of a practitioner is validation performance, the primary task of an optimization algorithm is to minimize training loss. Table 2 shows that Nesterov momentum achieves higher training accuracy than LARS, despite similar validation performance. Thus, it may be more appropriate to consider the layerwise normalization of LARS to be a regularization technique, rather than an optimization technique. This also explains its strong performance on ImageNet where regularization is critical (Bello et al., 2021).

Spending even more effort tuning LARS or Nesterov momentum would likely further improve the current state-of-the-art for that optimizer. Meaningful optimizer comparisons are only possible with independent and equally intensive tuning efforts, and we do not claim that either optimizer outperforms the other on this benchmark. That said, if the main evidence for LARS's utility as a "large-batch optimizer" is its performance on this particular benchmark, then more evidence is needed to quantify any benefit it has over traditional, generic optimizers like Nesterov momentum.

| $p_{\text{warmup}}$ | Nesterov | LARS |
|---|---|---|
| 1 | 75.79% | 75.97% |
| 2 | 75.92% | 75.69% |

| Optimizer | Train Acc | Test Acc |
|---|---|---|
| Nesterov | 78.97% | 75.93% |
| LARS | 78.07% | 75.97% |

Table 2: **(Left)** The best warmup schedule differs for Nesterov momentum and LARS. Values are medians over 50 training runs after setting $p_{\text{warmup}}$ without retuning other hyperparameters. **(Right)** Median train and test accuracies over 50 training runs for Nesterov momentum Configuration B and LARS.

## 2.2 LESSONS LEARNED

In hindsight, it was only necessary to make a few changes to the LARS pipeline to match its performance at batch size 32,768 with Nesterov momentum. However, Table 1 does not accurately represent the effort required when attempting to match a highly tuned training-speed benchmark.

Firstly, as described in Sections 2.1.2 and 2.1.3, the strong results of LARS depend partly on a few subtle optimization tricks and non-default values of uncommonly-tuned hyperparameters. Fortunately, in this case we could discover these tricks by examining the open-source MLPerf code, but machine learning research papers do not always report these important details. Researchers can easily waste experiments and produce misleading results before getting all of these details right. We demonstrate the importance of adding these tricks to our Nesterov momentum pipeline in Appendix C; without these tricks (or some new tricks), we likely would not have been able to match the LARS performance.

Secondly, the learning rate schedule really matters when trying to maximize performance with a relatively small step budget. Both LARS and Nesterov momentum are sensitive to small deviations from the optimized learning rate schedules in Figure 1, and neither schedule works as well for the other optimizer. Although relatively minor changes were sufficient to match LARS with Nesterov momentum, there is no way to know *a priori* how the optimal schedule will look for a new optimizer Wu et al. (2018). Even in toy settings where the optimal learning rate schedule can be derived, it does not fit into commonly used schedule families and depends strongly on the optimizer Zhang et al. (2019). Indeed, this problem applies to the other optimization hyperparameters as well: it is extremely difficult to know which are worth considering ahead of time. Finally, even when we narrowed down our hyperparemeter search spaces around the optimal point, the volume of our search spaces corresponding to near-peak performance was small, likely due to the small step budget (Shallue et al., 2019). We investigate how these effects change with a less stringent step budget in Section 4.

## 3 STRONGER BERT PRETRAINING SPEED BASELINES

You et al. (2019) developed the LAMB optimizer in the hope of speeding up training for BERT-Large (Bidirectional Encoder Representations from Transformers, Devlin et al., 2018). BERT training consists of two phases. The "pretraining" phase has two objectives: (1) predicting masked tokens based on the rest of the sequence (a masked language model), and (2) predicting whether two given sentences follow one from another. Finally, the "fine-tuning" phase refines the model for a downstream task of interest. BERT pretraining takes a considerable amount of time (up to 3 days on 16 Cloud TPU-v3 chips Jouppi et al. (2017)), whereas the fine-tuning phase is typically much faster. Model quality is typically assessed on the downstream metrics, not on pretraining loss, making BERT training a somewhat awkward benchmark for optimization research.

You et al. (2019) used LAMB for BERT pretraining with batch sizes up to 65,536 and claimed that LAMB outperforms Adam batch size 16,384 and beyond. The LAMB optimizer has since appeared in several NLP toolkits, including as Microsoft DeepSpeed and NVIDIA Multi-node BERT training, and as a benchmark task in MLPerf v0.7.[12]

As shown in Table 3, we trained Adam (with decoupled weight decay) baselines that achieve better results than both the LAMB and Adam results reported in You et al. (2019). Our new Adam baselines obtain better F1 scores on the development set of the SQuaD v1.1 task in the same number of training steps as LAMB for both batch size 32,768 and the hybrid 65,536-then-32,768 batch size training regime in You et al. (2019). We also ran Adam at batch size 65,536 to reach nearly the same F1 score as the hybrid batch size LAMB result, but in much fewer training steps. We believe 7,818 steps is a new state-of-the-art for BERT pretraining speed (in our experiments, it also improves upon the 76-minute record claimed in You et al., 2019). Additionally, at batch size 32,768 our Adam baseline got a better pretraining loss of 1.277 compared to LAMB's 1.342.

We used the same experimental setup as You et al. (2019), including two pretraining phases with max sequence lengths of 128 and then 512. In order to match You et al. (2019), we reported the F1 score on the downstream SQuaD v1.1 task as the target metric, although this metric introduces potential confounds: optimization efficiency should be measured on the training task using training and held-out data sets. Fortunately, in this case better pretraining performance correlated a with higher F1 score after fine-tuning. See Appendix B.2 for additional experiment details.

---

[12] We do not consider the MLPerf task in this paper since it is a warm-start, partial training task.

We tuned Adam hyperparameters independently for each pretraining phase, specifically learning rate $\eta$, $\beta_1$, $\beta_2$, the polynomial power for the learning rate warmup $p_{warmup}$, and weight decay $\lambda$, using quasi-random search (Bousquet et al., 2017). See Appendix E.2 for the search spaces.

In addition to hyperparmeter tuning, our improved Adam results at these batch sizes are also likely due to two implementation differences. First, the Adam implementation in You et al. (2019) comes from the BERT open source code base, in which Adam is missing the standard bias correction.[13] The Adam bias correction acts as an additional step size warm-up, thereby potentially improving the stability in the initial

| Batch size | Step budget | LAMB | Adam |
|------------|-------------|-------|-------|
| 32k | 15,625 | 91.48 | **91.58** |
| 65k/32k | 8,599 | 90.58 | **91.04** |
| 65k | 7,818 | – | **90.46** |

Table 3: Using Adam for pretraining exceeds the reported performance of LAMB in You et al. (2019) in terms of F1 score on the downstream SQuAD v1.1 task.

steps of training. Second, the BERT learning rate schedule had a discontinuity at the start of the decay phase due to the learning rate decay being incorrectly applied during warm-up [14] (see Figure 3 in Appendix B). This peculiarity is part of the official BERT release and is present in 3000+ copies of the BERT Training code on GitHub.

## 4 INVESTIGATING A LESS STRINGENT STEP BUDGET

Part of what makes comparing optimizers so difficult is that the hyperparameter tuning tends to dominate the comparisons (Choi et al., 2019). Moreover, tuning becomes especially difficult when we demand a fixed epoch budget even when dramatically increasing the batch size (Shallue et al., 2019). Fixing the epoch budget as the batch size increases is equivalent to demanding perfect scaling (i.e. that the number of training steps decreases by the same factor that the batch size is increased). We can view the role of hyperparameter tuning for large batch training as resisting the inevitable end of perfect scaling. For example, it might be possible to extend perfect scaling using delicately tuned learning rate schedules, but comparing optimizers under these conditions can make the learning rate schedule dominate the comparison by favoring some algorithms over others. Therefore, in order to better understand the behavior of LARS and LAMB compared to Nesterov Momentum and Adam, we ran additional ResNet-50 experiments with a more generous 6,000 step budget (vs 2,512 in Section 2) and a more simplistic cosine learning rate schedule. At batch size 32,768, this budget should let us reach better validation accuracy than the MLPerf target of 75.9%.

Although not mentioned in You et al. (2017), the state-of-the-art MLPerf pipeline for "LARS" actually uses both LARS and Heavy-ball Momentum, with Momentum applied to the batch normalization and ResNet bias parameters and LARS applied to the other parameters. You et al. (2019) does not mention whether LAMB was only applied to some parameters and not others. If layerwise normalization can be harmful for some model parameters, this is critical information for practitioners using LARS or LAMB, since it might not be obvious which optimizer to apply to which parameters. To investigate this, we trained both pure LARS and LAMB configurations, as well as configurations that did not apply layerwise normalization to the batch normalization and ResNet bias parameters. Moreover, LAMB's underlying Adam implementation defaults to $\epsilon = 10^{-6}$, rather than the typical $10^{-7}$ or $10^{-8}$. In some cases, $\epsilon$ can be a critical hyperparameter for Adam (Choi et al., 2019), so we included Adam configurations with both $\epsilon = 10^{-6}$ and $\epsilon = 10^{-8}$.

Table 4 shows the validation accuracy of these different configurations after training for 6,000 steps with batch size 32,768. In every case, we used a simple cosine decay learning rate schedule and tuned the initial learning rate and weight decay using quasi-random search. We used momentum parameters of 0.98 for Nesterov momentum and 0.929 for LARS, respectively, based on the tuned values from Section 2. We used default hyperparameters for Adam and LAMB except where specified. We set all other hyperparameters to the same values as the state-of-the-art LARS pipeline, except we set $\gamma_0 = 1.0$. See Appendix E.3 for more details. As expected, highly tuned learning rate schedules and optimizer hyperparameters are no longer necessary with a less stringent step budget. Multiple optimizer configurations in Table 4 exceed the MLPerf target accuracy of 75.9% at batch size 32,768 with minimal tuning. Training with larger batch sizes is *not* fundamentally unstable: stringent step budgets make hyperparameter tuning trickier.

---

[13] https://git.io/JtY8d  [14] See https://git.io/JtnQW and https://git.io/JtnQ8.

| Weights Optimizer | Bias/BN Optimizer | Top-1 |
|---|---|---|
| Nesterov | Nesterov | 76.7 |
| LARS | Momentum | 76.9 |
| LARS | LARS | 76.9 |
| Adam ($\epsilon = 10^{-8}$) | Adam ($\epsilon = 10^{-8}$) | 76.2 |
| Adam ($\epsilon = 10^{-6}$) | Adam ($\epsilon = 10^{-6}$) | 76.4 |
| LAMB | LAMB | 27.3 |
| LAMB | Adam ($\epsilon = 10^{-8}$) | 76.3 |
| LAMB | Adam ($\epsilon = 10^{-6}$) | 76.3 |

Table 4: Validation accuracy of ResNet-50 on ImageNet trained for 6,000 steps instead of 2,512. The second column is the optimizer that was applied to the batch norm and ResNet bias variables. We report the median top-1 accuracy over 5 seeds of the best hyperparameter setting in a refined search space. See Appendix E.3 for details.

In Table 4, "pure LAMB" performs extremely poorly: LAMB only obtains reasonable results when it is *not* used on the batch normalization and ResNet bias parameters, suggesting that layerwise normalization can indeed be harmful on some parameters. "Pure LARS" and Nesterov momentum perform roughly the same at this step budget, but the MLPerf LARS pipeline, which is tuned for a more stringent step budget, does not use LARS on all parameters, at least suggesting that the optimal choice could be budget-dependent.

Many new neural net optimizers, including LAMB, are introduced alongside claims that they do not require any—or at least minimal—tuning. Unfortunately, these claims require a lot of work to support, since they require trying the optimizer on new problems not used during the development of the algorithm. Although our experiments here are not sufficient to determine which optimizers are easiest to tune, experiments like those in Table 4 that operate outside the regime of highly tuned learning rate schedules can serve as a starting point, and we include preliminary "tunability" experiments using a simple grid search for LARS and Nesterov (details and more figures in Appendix D).

These grid search experiments sweep over 2 orders of magnitude of learning rate ($\eta$) and weight decay ($\lambda$), using 100 trials centered around the best settings from Table 4. Figure 2 shows the fraction of hyperparameter settings that reach a target validation accuracy. For very competitive accuracies ($> 76.6\%$), LARS reaches the target slightly more often than Nesterov, but otherwise the optimizers are similar or Nesterov more frequently reaches the target. Additionally, we analyze the best $\eta$ for each $\lambda$ in the grid search, and Nesterov appears to have an inverse relationship between $\eta$ and $\lambda$ while LARS has the benefit of an optimal $\eta$ independent of $\lambda$. This raises the question of if

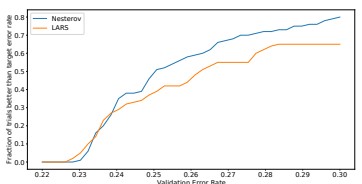

Figure 2: The fraction of hyperparameter settings that achieve better than a target validation error rate.

we could make Nesterov "easier" to tune by reparameterizing the search space to be $\eta$ by $\frac{\eta}{\lambda}$, similar to how LARS incorporates $\lambda$ into its step size denominator. Overall, LARS and LAMB do not appear to have an advantage in how easy they are to tune even on a dataset and model that were used in the development of both of those algorithms. LAMB is a variant of Adam and performs about the same as Adam with the same value of $\epsilon$ in Table 4; LARS is more analogous to Momentum and indeed Nesterov momentum and LARS have similar performance.

## 5 DISCUSSION

Our results show that standard, generic optimizers suffice for achieving strong results across batch sizes. Therefore, any research program to create new optimizers for training at larger batch sizes must start from the fact that Momentum, Adam, and likely other standard methods work fine at batch sizes as large as those considered in this paper. The LARS and LAMB update rules have no more to do with the batch size (or "large" batches) than the Momentum or Adam update rules. Although You et al. (2019) presented convergence rate bounds for LARS and LAMB to support their claims of superior performance, we show in Appendix A that Adam satisfies a similar bound to LAMB. These bounds all rely on very unrealistic assumptions.[15] Most of all, they are loose upper bounds on the worst case behavior of the algorithms, not accurate reflections of optimizer performance in reality. Whether layer-wise normalization can be useful for optimization or regularization remains an open

---

[15] All convergence bounds assume no momentum is used, and the $L_{avg}$ bound for LAMB also assumes $\beta_2 = 0$, when it is typically 0.999. Additionally, $L_{avg}$ could still be large if $L_\infty$ is large, but we leave an empirical analysis of this to future work.

question. However, if LARS and LAMB have any advantage over standard techniques, it is not that they work dramatically better on the tasks and batch sizes in You et al. (2017; 2019).

Our primary concern in this paper has been matching the state of the art—and establishing new baselines—for *training speed* measurements of the sort used to justify new techniques and algorithms for training with larger batch sizes. In contrast, many practitioners are more concerned with obtaining the best possible validation error with a somewhat flexible training time budget. Part of the reason why matching LARS at batch size 32,768 was non-trivial is because getting state of the art training speed requires several tricks and implementation details that are not often discussed. It was not obvious to us *a priori* which ones would prove crucial. These details do not involve changes to the optimizer, but they interact with the optimizer in a regime where all hyperparameters need to be well tuned to stay competitive, making it necessary to re-tune everything for a new optimizer.

In neural network optimization research, training loss is rarely discussed in detail and evaluation centers on validation/test performance since that is what practitioners care most about. However, although we shouldn't *only* consider training loss, it is counter-intuitive and counter-productive to elide a careful investigation of the actual objective of the optimizer. If a new optimizer achieves better test performance, but shows no speedup on training loss, then perhaps it is *not* a better optimizer so much as an indirect regularizer. Indeed, in our experiments we found that Nesterov momentum achieves noticeably better training accuracy on ResNet-50 than the LARS configuration we used, despite reaching roughly the same validation accuracy. Properly disentangling possible regularization benefits from optimization speed-ups is crucial if we are to understand neural network training, especially at larger batch sizes where we lose some of the regularization effect of gradient noise. Hypothetically, if the primary benefit of a training procedure is regularization, then it would be better to compare the method with other regularization baselines than other optimizers.

Ultimately, we only care about batch size to the extent that higher degrees of data parallelism lead to faster training. New optimizers have the potential to dramatically improve algorithmic efficiency across multiple workloads, but our results show that standard optimizers can match the performance of newer alternatives on the workloads we considered. Indeed, despite the legion of new update rule variants being proposed in the literature, standard Adam and Momentum remain the workhorses of practitioners and researchers alike, while independent empirical comparisons consistently find no clear winner when optimizers are compared across a variety of workloads (Schmidt et al., 2020). Meanwhile, as Choi et al. (2019) and our results underscore, comparisons between optimizers crucially depend on the effort spent tuning hyperparameters for each optimizer. Given these facts, we should regard with extreme caution studies claiming to show the superiority of one particular optimizer over others.

# 6   CONCLUSION

In this work, we demonstrated that standard optimizers, without any layer-wise normalization techniques, can match or exceed the large batch size results used to justify LARS and LAMB. Future work attempting to argue that a new algorithm is useful by comparing to baseline methods or results, including those established in this paper, faces a key challenge in showing that the gains are due to the new method and not merely due to better tuning or changes to the training pipeline (e.g. regularization tricks). Although gains from tuning will eventually saturate, we can, in principle, always invest more effort in tuning and potentially get better results for any optimizer. However, our goal should be developing optimizers that work better across many different workloads when taking into account the amount of additional tuning they require.

Moving forward, if we are to reliably make progress we need to rethink how we compare and evaluate new optimizers for neural network training. Given how sensitive optimizer performance is to the hyperparameter tuning protocol and how difficult it is to quantify hyperparameter tuning effort, we can't expect experiments with self-reported baselines to always lead to fair comparisons. Ideally, new training methods would be evaluated in a standardized competitive benchmark, where submitters of new optimizers do not have full knowledge of the evaluation workloads. Some efforts in this direction have started, for instance the MLCommons Algorithmic Efficiency Working Group[16], but more work needs to be done to produce incentives for the community to publish well-tuned baselines and to reward researchers that conduct the most rigorous empirical comparisons.

---

[16] https://mlcommons.org/en/groups/research-algorithms/

**Reproducibility Statement**

We will include a link to all code and all possible reproducibility instructions after the anonymized reviewing period is over. However, we are extremely detailed about our tuning procedures and dataset details, see Appendices B, E. We also include hosted Tensorboards for almost all of our tuning runs, available as `tensorboard.dev` links in the captions of the tables describing the hyperparameter searches. This allows readers to see, among other data, the exact points generated for each search in the "hparams" tab. All of this data can also be downloaded via the Tensorboard DataFrames API as CSV files or pd.DataFrames for future analysis.

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
