# OpenReview forum: "A Large Batch Optimizer Reality Check: Traditional, Generic Optimizers Suffice Across Batch Sizes"
_ICLR.cc/2022/Conference — ICLR 2022 Submitted_

### Official Review · Reviewer_F1kk · 2021-11-01

**Correctness:** 4
**Technical Novelty And Significance:** 1
**Empirical Novelty And Significance:** 2
**Recommendation:** 5
**Confidence:** 4

**Main Review:**

~~**A quick note in advance:** Apparently, the authors forgot to attach the appendix to the submitted version.
For my review, I didn't see a strong need to consult the appendix material,
which mostly contains additional explanation and experimental results.
So I won't consider this missing material problematic for this review,
but will conduct a final check of the appendix material once it has been
added during revision.~~

*Sorry for the erroneous comment on the appendix. I didn't realize that the appendix was in the supplements rather than attached to the main pdf (which was the case in other papers I have been reviewing).*

The paper is well-written. The experiment that have been conducted are
well-motivated and the experimental protocol is explained comprehensively and
clearly.
The authors do a good job explaining the lessons learned and make various
important points regarding the difficulty of farily evaluating and comparing
deep learning optimizers, e.g., the entanglement of optimization speed and
regularization effects.

I have one crucial criticism of this work. The paper puts significant effort in
the tuning of Nesterov/Adam to show that it can match the performance
achieved by LARS/LAMB. That answers one question: Is LARS/LAMB *necessary* to achieve
state-of-the-art results in large-batch optimization. While potentially eye-opening, this is a very limited
result. As the authors themselves write in the conclusion, one "can, in principle,
always invest more effort in tuning and potentially get better results
for any optimizer". A much more interesting and helpful question to ask would be:
Is LARS/LAMB *beneficial* in large-batch optimization? This would require to
assign a similar tuning protocol and budget to both methods. The hyperparameters
used for LARS are taken from published work (Kumar et al., 2019), which ostensibly
already put some effort into tuning those. However, a paper that critiques inadequate
comparisons in prior work should go the extra mile and guarantee an apples-to-apples
comparison. It seems quite likely that an equally well-tuned LARS/LAMB could see
some improvements.

**Summary Of The Paper:**

This paper takes a closer look at deep learning optimization methods tailored to
the large batch size regime, namely LARS and LAMB. It shows that generic optimizers
(SGD with Nesterov momentum and Adam, respectively) can achieve similar results in the large
batch regime given a careful tuning of their hyperparameters.

**Summary Of The Review:**

As stated above, the paper is well-written and clear. All claims are adequately
scoped and supported by the experiments provided in the paper.
This paper is not novel and it isn't trying to be. I expressly welcome papers
that double-check and critique prior work and try to disentangle effects with
carefully-designed experiments. However, I feel that the limited result of matching LARS/LAMB
instead of a full-fledged comparison of between LARS/LAMB and Nesterov/Adam greatly
diminishes the significance of the paper. With this shortcoming, in my opinion,
the paper narrowly misses the mark for publication at ICLR.

---

### Official Review · Reviewer_uA2C · 2021-11-02

**Correctness:** 3
**Technical Novelty And Significance:** 2
**Empirical Novelty And Significance:** Not applicable
**Recommendation:** 3
**Confidence:** 5

**Details Of Ethics Concerns:**

The key advantage of this paper is that the authors can use a significant amount of computing resources while other researchers can not. I worry that this paper may mislead the ML community and have a negative impact on academia's budget/planning.

**Main Review:**

The performance of LARS will be much higher if they use the same amount of computing resources to tune hyper-parameters.

For ImageNet, the authors did not report the results of a larger batch size (e.g. 64K or 128K). My experiments show that the performance of LARS would be much better than Nesterov for larger batch size.

With unlimited computing resources, the performance of LARS will still be better than Nesterov. The authors probably will eventually get results like Figure 7 of https://arxiv.org/pdf/2006.08517v1.pdf

If we take a look at Table 1, we can see that the hyper-parameters of Nesterov are meticulously selected or cherry-picked. Since the tuning cost is so high, it is almost impossible for users to consider this solution.

For example, to my knowledge, the authors of LARS did not tune the hyper-parameters of Batch Normalization. The authors of this paper used a lot of computing resources to tune the hyper-parameters of Batch Normalization.

"We ran a series of experiments, each of which searched over a hand-designed hyper-parameter search space using quasi-random search [Bousquet et al., 2017]."

To my knowledge, the LARS authors did not use any advanced hyper-parameter tuning methods. This means the comparison is not fair.

"Table 2 shows that Nesterov momentum achieves higher training accuracy than LARS, despite similar validation performance. Thus, it may be more appropriate to consider the layerwise normalization of LARS to be a regularization technique, rather than an optimization technique."

This conclusion is doubtful. Firstly, the authors probably need to sample many different convergence points (including low-accuracy points) to study of optimization pattern of LARS/Nesterov. Secondly, the generalization performance of the different minima may have different implicit properties, which can not be explained in such a simple way.

BTW, I can't reproduce the authors' results even though I totally trust the correctness of this paper. The reason is that we often need to use different hyper-parameters on different systems/hardware for large-batch training. A high tuning cost makes this process very hard.

For BERT, I suspect the authors tuned the hyper-parameters of fine-tuning process, which means the comparison is very unfair as the LAMB authors did not tune it.

Even if the authors did not tune the fine-tuning process, this comparison is still unfair because LAMB can work without tuning for changing batch size.

As mentioned by the LAMB authors (caption of Table 4 in https://openreview.net/pdf?id=Syx4wnEtvH): "We can achieve an even higher F1 score if we manually tune the hyperparameters"

The LAMB authors reported untuned results in Table 4 and Table 5 of https://openreview.net/pdf?id=Syx4wnEtvH

The authors of this paper not only tuned common hyper-parameters (e.g. learning rate) but also tuned some uncommon hyper-parameters (e.g. beta1, beta2 in Adam).

I suspect only big companies can do this kind of hyper-parameter searching for BERT per-training.

The novelty of this paper is very low. The key technical part is just tuning hyper-parameters. The authors also did not provide any deep analysis. If we don't need deep analysis, there are actually many blogposts on this topic (e.g. https://medium.com/fenwicks/tutorial-2-94-accuracy-on-cifar10-in-2-minutes-7b5aaecd9cdd). They can also significantly improve the performance of SGD with different optimization tricks. Similar tricks will likely improve the performance of LARS/LAMB.

The key advantage of this paper is that the authors can use a significant amount of computing resources while other researchers can not. I worry that this paper may mislead the ML community and have a negative impact on academia's budget/planning.

In my humble opinion, this paper is not a fit to top ML conferences like NeurIPS/ICML/ICLR.

**Summary Of The Paper:**

This paper was previously rejected by NeurIPS and ICML. I was one of the NeurIPS reviewers for this paper.
Although the authors made some changes, after carefully reading this paper, I found it did not change too much for key techniques and key contributions.
So I decided to use my previous NeurIPS review.

###############################################################################################################

The authors used a huge amount of computing resources to tune hyperparameters of Adam/SGD and claimed that they can match the performance of LARS/LAMB for large-batch training. I think the comparison is not fair.

**Summary Of The Review:**

The novelty of this paper is very low. The key technical part is just tuning hyper-parameters. The authors also did not provide any deep analysis. If we don't need deep analysis, there are actually many blogposts on this topic (e.g. https://medium.com/fenwicks/tutorial-2-94-accuracy-on-cifar10-in-2-minutes-7b5aaecd9cdd). They can also significantly improve the performance of SGD with different optimization tricks. Similar tricks will likely improve the performance of LARS/LAMB.

The key advantage of this paper is that the authors can use a significant amount of computing resources while other researchers can not. I worry that this paper may mislead the ML community and have a negative impact on academia's budget/planning.

In my humble opinion, this paper is not a fit to top ML conferences like NeurIPS/ICML/ICLR.

---

> ### Author Response · Authors · 2021-11-10
> **Copied review from previous conference, hopefully can have a discussion this time**
>
> We would like to note that this is an exact copy/paste of a review we received from a previous conference, which is unfortunate given that we added significant results (regarding hyperparameter sensitivity) based on our previous conference submission discussions with other reviewers. The reviewer who wrote this same exact review was the only one not to engage in our previous discussions, and seems to have ignored the thorough response we previously gave, which corrects what we believe are several incorrect or negligent statements in the review. We copy/paste our previous response below (and in another comment), in the hopes that we can have a discussion about it during this round:
>
> """
> Thank you for the review.
>
> The review makes multiple claims about unpublished results we do not have access to and do not have sufficient information to evaluate. We do not believe that we have any obligation to respond to those claims. We welcome future work to publish results (along with sufficient detail to reproduce them) that challenge our conclusions or improve on the baselines we provided.
>
> This review suggests on several occasions that we compared highly tuned optimizers (Nesterov momentum/Adam) to less tuned optimizers (LARS/LAMB). We would like to clarify that, as detailed in Sections 1 and 2, we instead compared Nesterov momentum to the highly tuned, state-of-the-art LARS numbers from the MLPerf training competition, which was tuned competitively by competition entrants for years. We agree that the results from the LARS paper were not competitive with the state of the art, so we did not compare to those and instead compared to the stronger MLPerf result. Regarding our comparison of Adam to LAMB, we only ran a single hyperparameter sweep for that workload. We agree that both Adam and LAMB could likely be improved with future tuning and we welcome future work to do so; our point is that a reasonably tuned Adam baseline is as good as the best published numbers of LAMB.
>
> Moreover, in Section 4 we ran an additional simpler comparison between Nesterov momentum and LARS, and between Adam and LAMB, without extensive tuning for any optimizer. We did not observe any significant improvement with LARS or LAMB over the generic counterpart.
>
> Regarding the concerns about the amount of resources required for us to test claims that standard optimizers can't match LARS/LAMB at large batch sizes: in order to compare to previous work, we were forced to use at least some of the same batch sizes. We also note that the total amount of computing resources that went into tuning LARS in the MLPerf training competition was almost certainly more than we used.
>
> The review claims without evidence that our hyperparameters were “meticulously selected or cherry-picked” and suggests that we used “advanced hyper-parameter tuning”. On the contrary, we simply used random search over the search ranges listed in the appendix. Unlike most papers, we documented every step of our tuning process, including intermediate experiments, which is above and beyond the standard in the literature. We welcome more detailed reviews of our search spaces to highlight any particular concerns.
>
> The reviewer’s suspicion that we tuned the BERT fine-tuning process is incorrect: we used the same fine-tuning procedure as the LAMB paper. Regarding the reviewer’s claim that the “comparison is still unfair because LAMB can work without tuning for changing batch size”, we would welcome more published results that compare the tunability of LAMB with other optimizers. However, it is difficult to perform such experiments fairly because one must, inevitably, select the search spaces for each optimizer. Moreover, our results in Section 4 do not support the claim that LAMB needs less tuning than Adam.
> """

---

> > ### Comment · Reviewer_uA2C · 2021-11-10
> > **Carefully reviewed but decided to keep the same**
> >
> > I actually carefully read the responses from the authors and appreciated their hard work.
> > I even tried my best to reproduce the results of this paper.
> > However, the authors did not address my concerns.
> > So I decided to keep my rating.

---

> ### Author Response · Authors · 2021-11-10
> **part 2 of our previous response**
>
> """
>
> Specific responses:
>
> - We disagree with the notion that it is uncommon to tune beta1 in Adam, or the equivalent momentum term in LARS, which is tuned in the paper the reviewer links (https://arxiv.org/abs/2006.08517 Table 4 shows multiple values of LARS momentum, the analogous parameter) and by the MLPerf Training competition (https://arxiv.org/abs/1909.09756 tunes beta1 and beta2 and also tunes the analogous LARS momentum parameter). Beta2 is slightly less common, but it is still tuned (along with beta1) in Adam in numerous papers, for example: https://arxiv.org/abs/1910.11758, https://arxiv.org/abs/2007.01547, https://arxiv.org/abs/1910.05446.
> -“For example, to my knowledge, the authors of LARS did not tune the hyper-parameters of Batch Normalization. The authors of this paper used a lot of computing resources to tune the hyper-parameters of Batch Normalization.” As we detail in Section 2.1, one of our settings of Nesterov momentum tuned Batch Normalization hyperparameters (Configuration A), but we also report a setting that achieves the 75.9% target without tuning these (Configuration B), so tuning Batch Normalization was not necessary in hindsight.
> -“BTW, I can't reproduce the authors' results even though I totally trust the correctness of this paper.” We will link to the exact code we used to run our experiments once the anonymous reviewing period is over. We wholeheartedly agree that reproducibility is a difficult issue in our field, as it took us six months to reproduce the LARS MLPerf ImageNet results in a new codebase.
> -The review references Figure 7 of https://arxiv.org/pdf/2006.08517v1.pdf, but this paper does not appear to tune any regularization terms, which we agree could lead to overfitting issues at larger batch sizes due to a lack of minibatch noise. We are yet to see any evidence that, as we increase the batch size, the regularization provided by minibatch noise cannot be supplanted by L2 or weight decay.
> """

---

> > ### Comment · Reviewer_uA2C · 2021-11-10
> > **Much higher tuning cost**
> >
> > 1. Adam and LAMB have the same number of hyper-parameters. Researchers could scale the batch size of BERT to 64K by LAMB without tuning beta1 or beta2;
> >
> > 2. It is very hard (or needs much a higher tuning cost) to achieve 75.9% without Configuration B
> >
> > 3. The 64K batch size performance of LARS-ImageNet has been reproduced by many researchers and engineers, for example:
> > https://arxiv.org/abs/1807.11205
> > http://learningsys.org/nips18/assets/papers/84CameraReadySubmissionYing_Kumar_Supercomputer.pdf
> > https://nnabla.org/paper/imagenet_in_224sec.pdf
> > https://arxiv.org/abs/1909.09756
> >
> > 4. Yes, eventually it will not work. But it can have a better performance with a reasonable tuning cost.

---

### Official Review · Reviewer_Fjz9 · 2021-11-03

**Correctness:** 4
**Technical Novelty And Significance:** 1
**Empirical Novelty And Significance:** 2
**Recommendation:** 6
**Confidence:** 4

**Main Review:**

Understanding large batch training is quite important in this days of large data, and setting up a proper baseline result is an important. And yet, my main concern is the novelty or significance of this work, in that the main message of this work is pretty much shared and revealed by previous works including Shallue et al. (2019), Zhang et al. (2019), Schmidt et al. (2020) and Choi et al. (2019): it highlights the importance of tuning the hyperparameters involved when comparing optimization algorithms; standard optimization algorithms are no worse in generalization at large batches. But unlike the previous works this work does not provide results for an extensive range of batch sizes (ok may it’s fine considering only large batch optimizers), any theoretical guarantee (I checked the proof but the convergence rate looks worse to me compared to that of LARS -- although I agree with you that it’s worst case bound, but still) or concrete evidence to support the idea of implicit regularization effects for small batch sizes.

Maybe a minor issue but I’d like to point out that the manuscript is quite unnecessarily lengthy; the main message is repeating, the details to secure evaluation validity, and the presentation of their results can all be more organized and cleaned up.

**Summary Of The Paper:**

This paper revisits the effectiveness of the optimizers designed for large-batch training such as LAMB and LARS by You et al. (2017, 2019) respectively. While it has been claimed and demonstrated (in perhaps limited settings though) that such optimizers can achieve better performance compared to other generic optimizers such as SGD or ADAM (in the sense that they don’t require a specific batch size), this paper re-evaluates these optimizers while fine-tuning all hyperparameters involved to potentially affect the result and finds that they do not work better as claimed; or, more precisely the standard optimization algorithms including Nesterov and Adam can match or outperform LARS as long as they are properly tuned. The paper provides empirical evidence obtained from Imagenet and BERT experiments to support their finding.

**Summary Of The Review:**

I appreciate the authors effort in correcting the common misconception that large batch training optimizers works better than standard optimizers through experiments, and I give a weak accept for now, but honestly I am not entirely certain if the paper contains substantial improvements or novelty compared to previous relevant works.

---

### Official Review · Reviewer_9Uka · 2021-11-08

**Correctness:** 4
**Technical Novelty And Significance:** 2
**Empirical Novelty And Significance:** 3
**Recommendation:** 5
**Confidence:** 4

**Main Review:**

The central message of the paper is expressed quite clearly and also backed up by substantial experiments, i.e., **LARS and LAMB don't seem to have a fundamental optimization benefit as opposed to simpler methods**. I do have some comments on the writing/results:

1) At many places in the paper the techniques are not fully explained, and their usage is taken for granted. This forces the reader to do a lot of back and forth between different papers to disambiguate and doesn't keep the work self-contained. It is ironic because the paper wants to promote rigor while specifying hyper-parameters. For instance, layerwise normalization is never formally written down in an equation. Similarly, how is L2 regularization performed? This is crucial to know how to interpret $\lambda$. What does the label smoothening coefficient $\tau$ do? What do you mean by decoupled weight decay with Adam?  All of this could have been avoided by using simple mathematical descriptions or through pseudocode.

2) LARS and LAMB were proposed to improve the batch size scaling while training. In all the experiments in the paper, the intent is to compare against them for a specific batch size that obtained the best performance on a benchmark task. How do the simpler methods compare against them in terms of the linear speed-up and critical batch sizes? Are they more or less scalable? Having a speed-up curve v/s batch size would have made the comparison more transparent as the authors claim that,
> The 88 epoch, 65,536 batch size result is faster in terms of wall-clock time but requires more training epochs, indicating that it is beyond LARS’s perfect scaling regime. Although LARS obtains diminishing returns when increasing the batch size from 32,768
to 65,536, future work could investigate whether Nesterov momentum drops off more or less rapidly than LARS.

    Also, how does robustness to hyper-parameter tuning change with the batch size (if it does)? Moreover, how do these behaviors vary across learning tasks (and not just Resnet with Imagenet)? These are important questions. After reading the paper, it was not clear to me which optimizer should I actually use for a given scale of the problem and systems constraints. As a paper, which tries to establish a new baseline this should have happened. Overall, the comparison was not as comprehensive as [this paper](https://arxiv.org/abs/1811.03600).


3) How did the authors come with the 75.9% threshold? It would have been more useful to give some second-moment statistics such as standard deviation or confidence intervals for hypothesis testing, instead of saying how many runs successfully crossed the threshold.

4) The paper takes credit for initiating discourse about things, which are already common knowledge. For instance,
>We show that future work must carefully disentangle regularization and optimization effects when comparing a
new optimizer to baselines.

    Any good paper should do that, and many bad papers don't understand that simple thing. This paper doesn't add anything new to that discourse. In table 2 the authors show that LARS has a slightly better test accuracy than SGD w/ momentum, but performs worse on optimization. There are many issues here.
    * First of all, how should I interpret these differences in performance, are they significant or not? What is the standard deviation across the different runs or how big are the confidence intervals for some p-value?
    * More importantly, how were the hyper-parameters tuned: to get the best validation performance or the best optimization performance? And in either case, how was this measured, through average performance at the tail of the optimization curve or just looking at a single number at the end?
    * If the hyper-parameters were indeed tuned using a validation data-set, how can the paper claim that the aim of the optimizers is to optimize sample loss? If I just cared about optimization, I would tune the hyper-parameters to minimize the optimization loss.
    * If I were to believe the assertion that layer-wise normalization should be interpreted as regularization, then what is the implicit bias it introduces to SGD? Can any differences in train and test performance now, be pushed under the rug as implicit regularization? Why couldn't this simply be explained in terms of the bias-variance trade-off/overfitting? Are there functional differences between the models learned through different optimizers, in terms of let's say prediction disagreement? How should I think about these issues in face of (less or more) overparameterization? These are all questions that are left unanswered. It seems the authors just want to show the marginal utility of LARS, instead of investigating what is happening under the surface.
    > Although the main concern of a practitioner is validation performance, the primary task of an optimization algorithm is to minimize training loss.
    * In light of the nuances I highlight above, this is not a correct statement. It depends on how the optimizer is sampling, tuning its hyper-parameters, etc.

5) **Hyper-parameters:** the warm-up schedule used for configuration B seems rather ad-hoc, have some other work used a quadratic warm-up before? Is there any rationale behind this?

Thus, I feel the paper can greatly improve by introducing more details rigorously, qualifying its statements, not overselling its contributions, and investigating some of its assertions further. Even then, I am not convinced that just showing that a previously accepted baseline, is not justifiably better is a big enough contribution. Specifically, it is unclear to me how useful is the entire line of work that tries to squeeze the most of data-parallel training for Resnets on Imagenet. There are other learning problems (like say in speech), and I don't understand how generalizable are any insights in this paper to neural network optimization.

**Summary Of The Paper:**

In data-parallel distributed training, increasing the batch size of the optimizer's updates is the most natural way to reduce wall-clock training time. Prevalent first-order stochastic methods such as SGD, provide a linear speed-up in the batch size, but only up to some critical batch size (c.f., [smooth convex optimization bounds](https://arxiv.org/abs/1106.4574), and [an empirical work suggesting the same](https://arxiv.org/abs/1811.03600)). This critical batch size is typically very different for different learning architectures and optimization algorithms, and it is often difficult to decouple the effect of,
* the bias-variance terms in optimization,
* improper or insufficient hyper-parameter tuning, and
* implicit regularization of the optimizer for extremely over-parameterized learning problems.

As a result, many existing works that propose new optimization algorithms, miss important baselines or don't compare against them fairly. This paper highlights such an issue with the recently proposed layerwise normalization techniques [LARS](https://arxiv.org/abs/1811.03600) and [LAMB](https://arxiv.org/abs/1904.00962), which build upon the update rules for SGD w/ Polyak momentum and Adam respectively. These methods were proposed to speed up the large batch (pre-)training of Imagenet and BERT respectively, and have gained a lot of attention in benchmark competitions. This paper underlines that these techniques improve optimization either marginally or not at all when compared to their vanilla first-order counterparts. Moreover, it highlights nuances in their hyper-parameter choices which are very important to consider. Most importantly, it establishes simpler baselines for improving future optimization algorithms for the considered learning tasks.

**Summary Of The Review:**

The paper does enough to justify to me that LARS and LAMB are not strictly better than simpler optimizers. But it doesn't do much more. There aren't many generalizable insights here. I highlight some areas where the paper can improve. I might be open to increasing my score if those concerns are addressed.

---

> ### Author Response · Authors · 2021-11-10
> **Response**
>
> We thank the reviewer for their time. We address several of their points below:
>
> - “layerwise normalization is never formally written down in an equation” while we assume a reader of this paper would be familiar with layerwise normalization, we had already written the update rules for LARS/LAMB in Algorithms 1, 2 in Appendix A
> -“Similarly, how is L2 regularization performed?” we use the phrase “L2 regularization” because we are performing L2 regularization in the traditional form, as opposed to weight decay or decoupled weight decay, and we assume a reader is familiar with L2 regularization.
> -“What does the label smoothening coefficient  do?” we assume a reader of this work is familiar with classic label smoothing, and we already include a citation to https://arxiv.org/abs/1512.00567
> -“ What do you mean by decoupled weight decay with Adam?” we do take for granted that a reader of this work is familiar with decoupled weight decay.
> -“How did the authors come with the 75.9% threshold?” at the beginning of Section 2 we introduce this as being from the popular MLPerf Training benchmark, “The MLPerf training benchmark for ResNet-50 v1.5 on ImageNet (Mattson et al., 2019) aims to reach 75.9% validation accuracy in the shortest possible wall-clock time.”
> -“It would have been more useful to give some second-moment statistics such as standard deviation or confidence intervals” while we do not provide the explicit stddev, you can refer to our figures in Appendices B, C to see candle plots of the distributions over seeds.
> -“‘We show that future work must carefully disentangle…’ Any good paper should do that, and many bad papers don't understand that simple thing. This paper doesn't add anything new to that discourse.” it is our opinion that many papers in the deep learning optimization community do not even acknowledge this discourse, which is why we highlighted it. We agree that we do not attempt to solve this discourse in this work.
> -“More importantly, how were the hyper-parameters tuned” we give thorough details about hyperparameter tuning in Appendix E, which we summarize and reference from the main text, “We ran a series of experiments, each of which searched over a hand-designed hyperparameter search space using quasi-random search (Bousquet et al., 2017). Between each experiment...are provided in Appendix E.4.”
> -“to get the best validation performance or the best optimization performance” we state multiple times in the text that we are tuning for the validation accuracy, “Specifically, we measure the median validation accuracy over 50 training runs with a fixed budget of 2,512 training steps at a batch size of 32,768”. While it is true that we tune on and then report the numbers on the standard ImageNet “validation” (we do not introduce a separately held-out validation set), this is an unfortunately common practice in the deep learning optimization community that we are following in order to have a fair comparison to the LARS MLPerf numbers.
> -“And in either case, how was this measured, through average performance at the tail of the optimization curve or just looking at a single number at the end?” we do not believe it is common to report average performance at the tail of the optimization curve, at least in the deep learning optimization community. Our metrics consider if a training run reached 75.9% validation accuracy within the first 2512 steps of training, as we describe in the text.
> -“If I just cared about optimization, I would tune the hyper-parameters to minimize the optimization loss.” we agree, but in real-world systems practitioners (who are an intended audience of our work) rarely care about training loss and aim to optimize validation metrics.
> -“what is the implicit bias it introduces to SGD? Can any differences in train and test performance now, be pushed under the rug as implicit regularization? Why couldn't this simply be explained in terms of the bias-variance trade-off/overfitting?” we are unsure what the reviewer means by this, we do not explicitly prove that layerwise normalization implicitly regularizes the model, because this paper is not a theoretical analysis of layerwise normalization but instead an introduction of baselines for large batch training.
> -“Are there functional differences between the models learned through different optimizers, in terms of let's say prediction disagreement?” we believe this is beyond the scope of this work.
> -“It seems the authors just want to show the marginal utility of LARS, instead of investigating what is happening under the surface.” what would the reviewer propose we do to investigate this (besides look at model disagreement, which would not reveal any fundamental mechanics of layerwise normalization)?

---

> > ### Comment · Reviewer_9Uka · 2021-11-11
> > **Response**
> >
> > I'd first address the first four bullet points. I strongly believe that these details can be clarified further **in the paper itself**. I would suggest adding the pseudocode to the main paper. The authors spend a lot of time talking about philosophical issues (which are admittedly important) in the paper, but as I said before, I don't think they take the discourse here forward. They can gain more space by cutting down here, and including precise details about what they mean when they use a technical term (which could have a couple of meanings). I also don't believe it is an appropriate response to say that something is "well known" in the deep learning optimization community, and should thus not be evaluated with the same level of scrutiny as any other paper in optimization. To clarify, I don't think that the authors missed specifying something, which was detrimental to me understanding the message of the paper, but I do believe that they could be more formal and specific in certain places.
> >
> > My second point has not been addressed at all.
> >
> > > at the beginning of Section 2 we introduce this as being from the popular MLPerf Training benchmark
> >
> > This is an appeal to authority fallacy. I am asking why the benchmark chose this threshold. There must be a statistical reason for doing so. I am asking you to mention that reason **in the paper**.
> >
> > > While it is true that we tune on and then report the numbers on the standard ImageNet “validation” (we do not introduce a separately held-out validation set),
> >
> > I am not sure I understand, I would appreciate it if you could elaborate. Are you saying there are two data sets and not three? In any case, I think it is incorrect to first tune your model to perform well on a validation set, and then claim it has a superior training performance (which should anyways be the goal of an optimizer). That is changing the goal post. Even if you don't report performance on a held-out data-set, it is still wrong to hope that the optimizer should perform better on training loss, if it has been tuned to do so on the validation loss. This becomes important because you claim the main job of an optimizer is to reduce training loss, which is true if it is also tuned to do so.
> >
> > > we agree, but in real-world systems practitioners (who are an intended audience of our work) rarely care about training loss and aim to optimize validation metrics.
> >
> > Ok, but then do not claim the main job of an optimizer is to minimize training loss, and claim that layerwise normalization is actually a regularization technique. I hope you see the fallacy here given your tuning setup.
> >
> > > we are unsure what the reviewer means by this, we do not explicitly prove that layerwise normalization implicitly regularizes the model, because this paper is not a theoretical analysis of layerwise normalization but instead an introduction of baselines for large batch training.
> >
> > It is completely fine, that you don't prove it explicitly. I am just not convinced that your **empirical evidence** also backs this assertion. Which is why I said, it needs further proof. Moreover, if you actually think the paper is about just introducing baselines (which are actually just baselines past work should have considered but it didn't), then please don't make assertions in the paper, which are not going to be investigated further.
> >
> > > “It seems the authors just want to show the marginal utility of LARS, instead of investigating what is happening under the surface.” what would the reviewer propose we do to investigate this (besides look at model disagreement, which would not reveal any fundamental mechanics of layerwise normalization)?
> >
> > I don't know, but I am convinced LARS is perhaps not the optimal baseline to compare against (given a simpler optimizer performs very similarly). As I said before, I think this paper makes an important contribution in establishing this, but I don't think it has enough technical novelty. It would have added to the paper, if the underlying mechanics of layerwise normalization could b explored further.

---

### Author Response · Authors · 2021-11-10
**Thank you for your reviews**

Overall there appears to be a common theme amongst the more positive reviews that introducing baselines that match or outperform newer methods is not novel or noteworthy enough for a conference paper. We believe this may be an irreconcilable difference in opinion.

---

### Decision · Program_Chairs · 2022-01-20

**Decision:**

Reject

**Comment:**

This paper experimentally shows that the commonly used standard solvers such as Nesterov momentum or Adam can achieve the same performance as optimizers such as LARS and LAMB specially proposed for large batches training.

Large batch training is a very important topic, and if the author's argument is true, it might be an interesting discovery.

However, all reviewers were concerned about its limited technical contribution. Not only that, but above all, when tuning the optimizer using the same computational resource (for a new task), it seems unclear whether the standard optimizer can achieve as much performance as large batch optimizers (currently they tune the standard optimizers, fixing the hyperparameter for large batch solvers to match their performances). The authors did not answer the reviewer's questions about it, and they did not answer the reviewer's other questions in great detail. Through discussion among reviewers, all reviews agreed on this concern and agreed to reject this paper.

The quality of the paper will be greatly improved if this concern is resolved.